# Parent-Integrated Interventions to Improve Language Development in Children Born Very Preterm

**DOI:** 10.3390/children10060953

**Published:** 2023-05-27

**Authors:** Anne Synnes, Thuy Mai Luu, Jehier Afifi, May Khairy, Cecilia de Cabo, Diane Moddemann, Leonora Hendson, Amber Reichert, Kevin Coughlin, Kim Anh Nguyen, Lindsay L. Richter, Fabiana Bacchini, Khalid Aziz

**Affiliations:** 1Department of Pediatrics, University of British Columbia, Vancouver, BC V6T 1Z3, Canada; lrichter@bcchr.ca; 2BC Women’s Hospital and Health Centre, Vancouver, BC V5Z 4H4, Canada; 3BC Children’s Hospital Research Institute, Vancouver, BC V5Z 4H4, Canada; 4Department of Pediatrics and Research Center, Centre Hospitalier Universitaire Sainte-Justine, Université de Montréal, Montreal, QC H3T 1C5, Canada; thuy.mai.luu@umontreal.ca; 5Department of Pediatrics, Dalhousie University, Halifax, NS B3H 4R2, Canada; jehier.afifi@iwk.nshealth.ca; 6Division of Neonatal Perinatal Medicine, IWK Health Centre, Halifax, NS B3K 6R8, Canada; 7Department of Pediatrics, Montreal Children’s Hospital, McGill University Health Centre, Montreal, QC H4A 3J1, Canada; may.khairy.med@ssss.gouv.qc.ca; 8Department of Paediatrics and Child Health, University of Manitoba, Winnipeg, MB R3A 1S1, Canada; cdecabo@hsc.mb.ca (C.d.C.); dmoddemann@hsc.mb.ca (D.M.); 9Department of Pediatrics, Alberta Children’s Hospital, University of Calgary, Calgary, AB T3B 6A8, Canada; leonora.hendson@albertahealthservices.ca; 10Glenrose Rehabilitation Hospital, Edmonton, AB T5G 0B7, Canada; amber.reichert@albertahealthservices.ca; 11Children’s Hospital at London Health Sciences Centre, London, ON N6A 5W9, Canada; kevin.coughlin@lhsc.on.ca; 12Jewish General Hospital, Montreal, QC H3T 1E2, Canada; kim-anh.nguyen2@mcgill.ca; 13Canadian Premature Babies Foundation, Etobicoke, ON M8X 1Y3, Canada; fabiana@cpbf-fbpc.org; 14The Office of Lifelong Learning, University of Alberta, Edmonton, AB T6G 1C9, Canada; khalid.aziz@ualberta.ca

**Keywords:** prematurity, patient-oriented research, family integrated care, parent perspectives, quality improvement

## Abstract

Neurodevelopmental challenges in children born very preterm are common and not improving. This study tested the feasibility of using Evidence-based Practice to Improve Quality (EPIQ), a proven quality improvement technique that incorporates scientific evidence to target improving language abilities in very preterm populations in 10 Canadian neonatal follow-up programs. Feasibility was defined as at least 70% of sites completing four intervention cycles and 75% of cycles meeting targeted aims. Systematic reviews were reviewed and performed, an online quality improvement educational tool was developed, multidisciplinary teams that included parents were created and trained, and sites provided virtual support to implement and audit locally at least four intervention cycles of approximately 6 months in duration. Eight of ten sites implemented at least four intervention cycles. Of the 48 cycles completed, audits showed 41 (85%) met their aim. Though COVID-19 was a barrier, parent involvement, champions, and institutional support facilitated success. EPIQ is a feasible quality improvement methodology to implement family-integrated evidence-informed interventions to support language interventions in neonatal follow-up programs. Further studies are required to identify potential benefits of service outcomes, patients, and families and to evaluate sustainability.

## 1. Introduction

Children born very preterm, at fewer than 29 weeks gestation, face several neurodevelopmental challenges, which are both common and impactful on their lives. Despite improvements in survival and neonatal outcomes after extremely and very preterm birth, improvements in rates of cognitive and language difficulties have been minimal [1]. Prematurity is not the sole risk factor for adverse neurodevelopmental outcomes per se [2]. Brain magnetic resonance imaging studies suggest that insults during pregnancy, birth, the neonatal period, and early childhood when the preterm brain is particularly vulnerable, can lead to abnormalities of brain maturation and white matter injury [3,4,5] and may be preventable. Language and cognition are important aspects of a child’s development and social well-being. Language and cognitive delays, defined as Bayley Scales of Infant and Toddler Development 3rd edition (Bayley-III) [6] composite scores less than 85 (less than 1 standard deviation) occur in more than one-third of very preterm children in Canada [7]. Cognitive and language development are influenced by parental education and environment [8]. Brain plasticity, driven by active synaptogenesis and preterm brain maturation in the first two years after birth, provides opportunities to positively influence cognitive and language outcomes [9].

Systematic reviews and meta-reviews [9,10,11,12] have identified interventions that improve infant developmental outcomes in children born preterm. Spittle et al.’s systematic review of randomized control trials showed improved cognitive outcomes in preterm populations with early developmental interventions post-hospital discharge [9]. The interventions varied in intensity, setting, timing, and approach, with those incorporating parent–infant interactions showing the most promise [10]. Other effective programs have been evaluated in term-born populations. The Reach Out and Read Program effectively improved receptive and expressive language outcomes, especially for children in socially disadvantaged homes [13]. Child development is a result of a complex interaction between biological, medical, and environmental factors [10], and the systematic reviews do not identify any simple intervention that can be feasibly implemented to address developmental delays in children born preterm.

The Canadian Neonatal Network^TM^ developed Evidence-based Practice to Improve Quality (EPIQ), a continuous quality improvement methodology that uses the best available evidence to identify site-specific targets, implement change, and evaluate outcomes [14]. EPIQ uses local data to direct site-specific and effective interventions derived from systematic reviews and the evidence-based literature and uses a network of experts and quality improvement techniques, such as «Plan-Do-Study-Act cycles», to improve outcomes. Using EPIQ in neonatal intensive care units, the Canadian Neonatal Network effectively and sustainably reduced the incidence of short-term neonatal morbidities [15,16,17]. 

Although most neonatal follow-up programs have traditionally focused on screening children for risk of developmental impairment, their multidisciplinary teams and expertise do equip them with the skills and potential resources to provide simple interventions to improve the development of the preterm child. To the best of our knowledge, the interventions described in the systematic reviews to improve neurodevelopmental outcomes have not been implemented in Canadian neonatal follow-up programs. The aim of this study was to test the feasibility of implementing changes using the EPIQ approach to improve language or cognitive outcomes in children born very preterm in Canadian neonatal follow-up programs. As parents and caregivers play a vital role in supporting their child’s development, we integrated parents into the implementation process for this Parent-EPIQ study.

We expected that participating sites would implement four to six intervention cycles and that at least 75% of the audits performed during each cycle would meet or surpass the identified target.

## 2. Materials and Methods

### 2.1. Design and Setting

This prospective study, coordinated by the Canadian Neonatal Follow-Up Network, was performed at 10 of 26 Canadian neonatal follow-up programs between 2017 and 2022. Research ethics approval was granted by the coordinating site at the University of British Columbia Children’s and Women’s Research Ethics Board (H17-00573) and at all participating sites.

The Parent-EPIQ study was implemented over three phases: a preparatory information-gathering phase, a training collaboration phase, and an implementation phase. In the preparatory phase, evidence-based interventions that improve cognitive or language development were identified. Spittle’s systematic review [9] and Vanderveen’s meta-analysis [10] were sources for cognitive outcome interventions. For language outcomes, early interventions to enhance language and social development with a specific focus on communication and parent–infant interactions were identified in a systematic review. Methodological quality was assessed using the Cochrane collaboration tool for risk of bias, and studies of good quality (score of ≥5) were selected [18,19,20,21,22,23,24,25,26,27,28,29] (Appendix A).

Simultaneously, an online EPIQ education tool was adapted for the Parent-EPIQ study by a co-investigator (KA), who had previously played a leading role in developing the EPIQ methodology. EPIQ uses 3- to 6-month Plan–Do–Study–Act cycles to implement local changes. EPIQ has 10 steps with tools to aid users at each step. EPIQ follows a design-thinking methodology, sequencing the 10 steps into (a) understanding the improvement opportunities (steps 1–3: five whys, force field analysis, and fishbone), (b) deciding how to address them (steps 4–6: feasibility tool, process mapping, and SMART indicators), and (c) engaging and acting on that decision (steps 6–10: EPIQ aim form, engagement tool, EPIQ change form, and run charts). EPIQ allows each site to individualize the process to its own situation. The design of the curriculum, using social constructivism [30,31], ensures that quality improvement-naïve participants, such as high school graduate parents, can actively participate alongside health care practitioners with or without quality improvement expertise. The 10 steps outline the progress of a simulated or real-life quality improvement project from its inception (a change idea) to its execution (Plan–Do–Study–Act) and its results. Following the 10 EPIQ steps satisfies many components of the Revised Standards for Quality Improvement Reporting Excellence (SQUIRE 2.0). Derived from the workshop presentations, videos reviewing the 10 steps were created for virtual, asynchronous learning.

During the training phase, a lead health care professional at each participating site created a Parent-EPIQ team of six to eight members with experience, skills, and/or influence related to the target outcome, including neonatal follow-up multidisciplinary health care professionals, one or more parent representatives, and other clinical or community representatives, as required. Site investigators recruited parents in several ways: parents expressed interest, parent advisory committee suggestions, and results from surveys of parent interests. Participation of all team members, including parent partners, was voluntary and involved attending team meetings and reading communications. Parent-EPIQ training videos representing three to four steps in the EPIQ process were circulated to site investigators prior to monthly teleconferences between September 2017 and February 2018. In the teleconferences, the content of the videos was reviewed with a content expert and facilitators. Local Parent-EPIQ teams met locally thereafter to practice using the EPIQ steps and were encouraged to use real quality improvement change ideas during this learning phase. Any evidence-based changes to improve language and/or cognitive outcomes prior to 18 months of age in the preterm population assessed by the site and within the sphere of influence of the Parent-EPIQ team were considered to be within the scope of this study. Site-specific baseline incidence rates of language and cognitive delays at 18 months corrected age were circulated. This was possible since all sites had previously submitted data to the Canadian Neonatal Follow-Up Network, including cognitive and language abilities assessed using the Bayley-III [7] (Figure 1).

Finally, during the intervention phase, each participating Parent-EPIQ site implemented four to seven PDSA cycles over the study period. Parent-EPIQ sites selected the interventions based on their local data and submitted EPIQ aim forms, process indicators, and audit results to the coordinating site. Teleconferences with experts were held quarterly with sites during the implementation phase to review progress, ensure feasibility and fidelity to the EPIQ methodology, and provide rapid fertilization of knowledge translation ideas. 

### 2.2. Data Collection and Analyses

The descriptive variables were obtained from the electronically submitted EPIQ aim forms, process indicators, and audit results and confirmed with site investigators. 

The two feasibility criteria were defined as: (1) 70% of participating sites complete at least four intervention cycles during the study and (2) 75% of all intervention cycles meet or surpass their target goals. At each site, for each EPIQ cycle, an aim form with specific measurable goals was completed and an audit was performed to address whether targets were met. The number of cycles completed for each site was recorded and the percentage of sites completing at least four intervention cycles was calculated. Site investigators who were unable to meet the target were asked to identify the factor(s) which hindered the completion of at least four cycles. The percent of audit results that met or surpassed target goals was calculated for all cycles at all sites.

Using the conceptual framework of Cane et al. [32], a survey was created to assess the barriers and facilitators to the implementation of Parent-EPIQ and circulated to participating site investigators by email after the completion of all the intervention cycles in June 2022.

## 3. Results

### 3.1. Site Description and Participation

Twelve Canadian neonatal follow-up programs expressed initial interest and 10 sites obtained local research ethics board approval and participated. Site size was described as large, medium, or small based on the number of participants with 18-month corrected age data uploaded from all participating Canadian neonatal follow-up programs in 2019 (Table 1).

### 3.2. Number of Parent-EPIQ Intervention Cycles Completed

Eight of ten sites completed four or more cycles (Figure 2), with one site completing two cycles, one site completing three, three sites completing four, one site completing five, two sites completing six, and two sites completing seven cycles. The planned two-year intervention phase was interrupted by the COVID-19 pandemic. As a result, the intervention phase was extended for one year. Site investigators at the two sites that were unable to complete four cycles reported the COVID-19 pandemic and staffing issues as reasons for an incompletion. 

### 3.3. Number of Parent-EPIQ Intervention Cycles That Met Goals

Forty-eight cycles were completed. All sites and all cycles targeted language improvement since language delay rates were higher than cognitive delay rates at all sites (Figure 1), and the evidence and feasibility of implementing interventions to improve language and communication were deemed easier than targeting cognitive outcomes. Intervention cycles are summarized in Table 1.

Audits were performed by measuring the percentage achieved against a target or completion of a task. Qualitative input was also sought to interpret the results or guide future intervention cycles. A review of the audits showed that aims were met for 41 of the 48 cycles (85%).

### 3.4. Description of Parent-EPIQ Intervention Cycles

Each site used the ten EPIQ steps to understand the improvement opportunities at their own site, how to address them with their own resources, and how to engage their own members to create interventions tailored to their own situation. Sites also had access to scientific evidence from the literature and systematic reviews and the experiences of colleagues at other sites to guide their choices. The implemented interventions are shown in Table 1. At most sites, interventions in the first cycle involved educating either neonatal follow-up program staff (n = 8) or parents (n = 1). The most common interventions thereafter targeted increasing parent communication with their child (ren) by reading, talking, or singing to their child either in the neonatal intensive care unit or after discharge to home. Other interventions created educational tools or targeted improving parent–infant interactions, parents’ mental health, or infant music therapy (Table 1). The setting for 21/48 (44%) of cycles was the neonatal follow-up programs, 24/48 (50%) involved the neonatal intensive care unit, and the remaining three cycles involved both or another setting.

### 3.5. Reasons That Parent-EPIQ Intervention Cycle Goals Were Not Met

The reasons for not achieving goals varied. Barriers to health care professionals attending educational sessions were not appreciated at three sites (site 1, cycle 1; site 2, cycle 2; and site 8, cycle 1). At site 7, cycle 3, the aim was to implement a systematic screening for post-partum depression. One health care professional had valid reasons for not agreeing with the project. As a result, after the completion of this study, a clinical pathway for parents with post-partum depression was developed. The team learned that a step to raise awareness among all staff members and obtain 100% agreement with the protocol was a necessary interim step. At site 2, cycle 5, an increase from 56% to 64% in the desired number of parents talking, reading, or singing during skin-to-skin care was observed, but the 70% target was not reached, and more time may have been needed to reach the goal. Site 7, cycle 5 was interrupted by the COVID-19 pandemic-related restrictions. For the remaining cycles, mostly the last cycle in the study, reasons for not achieving the targets were not identified.

### 3.6. Barriers and Facilitators

Responses to the barriers and facilitator questionnaire from nine of the ten sites are shown in Table 2. All respondents felt they had the required knowledge and understanding of the EPIQ process. Most sites (89%) had team members who attended the EPIQ in-person workshop as well as the virtual teaching, and 86% of those who attended the workshop felt it was essential. Over half of the participants felt they were able to find evidence to support their intervention cycles and found the systematic reviews and support from the teleconferences to be helpful. A team member, identified as a champion, and institutional support were important facilitators. The COVID-19 pandemic was the major barrier identified.

## 4. Discussion

Providing the child born very preterm with the best possible future is a shared goal of health care providers in neonatal follow-up programs, neonatal intensive care units, parents, families, and society in general. In this study, we demonstrated how the Parent-EPIQ method can be used in a neonatal follow-up program setting to implement evidence-based parent-integrated interventions. Our study introduced two novel ideas to address the high incidence of language delays in the preterm population: integration of parent partners in the process and adaptation of the EPIQ process to target neurodevelopmental outcomes in the neonatal follow-up program ambulatory care setting.

Parent-EPIQ teams had one to two parents on each team, with relatively low turnover during the study. Parents provided a different lens as to whether interventions would be acceptable and feasible for families. For example, parent participants identified that at the first neonatal follow-up program infant visit, parents are physically and emotionally overwhelmed. In contrast, most parents spend many hours in the neonatal intensive care unit wanting to be engaged and involved. Parents also brought a variety of skills to the teams. Compensation for travel, babysitting, and time commitments was offered but declined in most cases, reflecting parent willingness to contribute. Involving parents in new health care interventions is innovative and has not been previously reported, and our study has shown this to be both possible and beneficial.

Several other adaptations were made to the EPIQ process in this study. EPIQ was taught asynchronously using video modules and teleconference support. The feedback showed that site investigators felt the need to attend in-person workshops in addition to the video modules to be comfortable facilitating their teams. During the COVID-19 pandemic, synchronous virtual meetings became common, accepted, and improved. Our findings that in-person workshops are necessary may not apply to current virtual educational platforms. 

In Parent-EPIQ, the teams worked in neonatal follow-up program ambulatory care settings, which differ from neonatal intensive care unit settings. The neonatal follow-up program multidisciplinary health care professionals have different time constraints with outpatient visits and other commitments than neonatal intensive care unit staff. Fortunately, most neonatal follow-up programs function as multidisciplinary teams and adapt quickly to the Parent-EPIQ model.

The aims of this study were built on the pyramid of drivers shown in Figure 3. In addition to the fundamental EPIQ components (train teams to use EPIQ; use local data and systematic reviews to create evidence-based bundles, education, and awareness), the Parent-EPIQ model added parental integration and created engagement within a community of sites. Though EPIQ can be used at individual sites, it is more effective when used within a community with shared interests [16].

In keeping with the EPIQ model, each site had access to their local data on cognitive and language delay in children born upon fewer than 29 weeks gestation, which was provided by the coordinating center. (Figure 1). In this study, searches of the literature relevant to the overarching aim were reviewed and performed by the coordinating site, which facilitated the identification of improvement opportunities but still allowed sites to individually design intervention cycles appropriate for their own situation. The sharing of ideas and materials at the regular teleconferences allowed for learning from others while maintaining the flexibility to try new ideas. Most sites targeted an educational activity in the first cycle (Table 1). Ideas, such as collaborating with public libraries, spread from one site to another. Some interventions, such as music therapy, were unique due to locally available resources. As expected, the EPIQ process resulted in different interventions at different sites. The aim was to evaluate the EPIQ process and not to create standardized interventions.

Interventions could all be traced to the 10-step EPIQ process. Lack of knowledge of language development was identified by all sites as a barrier and became a natural first step. Local academic resources [33] and expertise were used to enhance knowledge and also influenced subsequent cycles. For example, the scientific evidence that early language development starts in utero [33,34] guided sites to consider earlier-in-life interventions. For sites with close links to their neonatal intensive care unit, interventions were started in the neonatal intensive care unit. This shift was reinforced by parents who identified their preference for neonatal intensive care unit interventions. 

Quality improvement interventions should be based on the best available evidence. The Parent-EPIQ interventions were guided by systematic reviews ([9,10], and Appendix A). Randomized controlled trials using the Mother–Infant Transaction Program [18] or modified versions [35] and the Infant Behavioral Assessment and Intervention Program [23] have shown improvement in cognitive outcomes up to school age [35,36]. In these complex programs, the goals are to help parents recognize and understand their baby’s behavioral cues and promote responsive and positive nurturing by modulating the environment, adapting everyday care, and enhancing interactions. Though it was not feasible to implement any of the programs identified in the systematic review, the theory behind these programs influenced interventions, such as parent teaching in the neonatal intensive care unit. Maternal depression affects infant language development by altering mother–infant interactions [16,37,38]. One site, therefore, targeted screening for maternal depression.

Despite the successes, two sites were not able to complete the goal of four intervention cycles. Parent-EPIQ requires a motivated and engaged team with a leader or champion to meet; review goals, data, and evidence; and implement a plan. Characteristics of neonatal intensive care units that facilitate making changes include staffing issues, consistency in practice, the approval process, a multidisciplinary approach to care, frequency and consistency of communication, the rationale for change, and the feedback process [39,40]. Institutions can therefore create environments that support quality improvement and change. As is often the case, to succeed requires a sustained commitment. 

Though all 26 Canadian neonatal follow-up programs were invited to join the study, participation was voluntary, and 16 sites, mostly smaller ones with fewer resources, elected not to. We had previously identified the considerable variability in the size and available resources of Canadian neonatal follow-up programs [41] and their ability and academic expectations to participate in research. The experiences learned in this study will help sites evaluate whether and what would be needed to participate in future implementation projects.

On the other hand, not meeting some EPIQ intervention cycle aims is expected and can be a learning opportunity. The team may identify previously unrecognized barriers, which can subsequently be remediated and facilitate success in the next cycle. 

In this study, our overarching aim was to improve language or cognitive development, outcomes which can only be evaluated months or years after the intervention. We, therefore, evaluated the ability of sites to implement the Parent-EPIQ process and created a smart aim of 75% success in meeting goals. 

In implementation science frameworks [42], implementation outcomes such as acceptability, adoption, appropriateness, and feasibility are necessary before improvements are seen in service and patient outcomes. Our study focused on implementation outcomes. We demonstrated that Parent-EPIQ can be implemented in many but not all Canadian neonatal follow-up programs and explored the barriers and facilitators to success. This study looked at a limited number of measures of the feasibility of Parent-EPIQ interventions and did not assess the effect on cognitive and language outcomes or other measures. It is an exploratory study that sets the stage for further implementation research in this area.

Improving outcomes of children born preterm is important but has been challenging. Considering the many variables that may affect the long-term outcomes of children born preterm, multiple strategies to support development must be implemented at many stages during early life. EPIQ has improved neonatal intensive care unit care practices in Canada [15,16,17] with a resultant reduction in neonatal morbidities associated with adverse neurodevelopmental outcomes compared to other similarly well-resourced health systems [43]. This needs to continue. Our study suggests that many neonatal follow-up programs have the human resources and the ability to implement support for the preterm population as well as provide individual screening and surveillance. Further study is required to understand if and how Parent-EPIQ can be implemented in all neonatal follow-up programs. 

## 5. Conclusions

Parent-EPIQ is a feasible quality improvement methodology to implement family-integrated evidence-informed interventions to engage parents and support language interventions for babies in neonatal follow-up programs. Parents are willing to participate and provide meaningful benefits to the team. The Parent-EPIQ process brings many advantages, such as simple steps, team cohesion, collaboration with other teams, goal-setting, empowerment, and social constructivism. Site collaboration catalyzes and supports activities. Evidence-informed best practices can be applied. Long-term outcomes can be replaced by surrogate short- and medium-term processes. Using EPIQ facilitates reporting using SQUIRE 2.0. Further studies are required to identify the benefits on service outcomes, patients and families, and the sustainability of Parent-EPIQ.

## Figures and Tables

**Figure 1 children-10-00953-f001:**
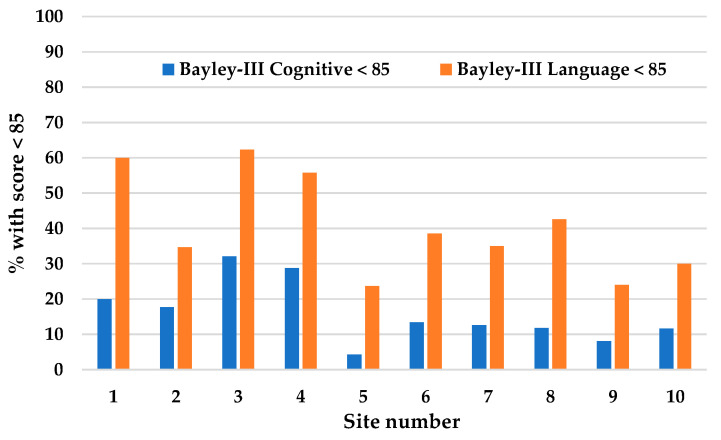
Percent of very preterm children (<29 weeks gestation) with cognitive and language delays at 18 months corrected age at participating sites.

**Figure 2 children-10-00953-f002:**
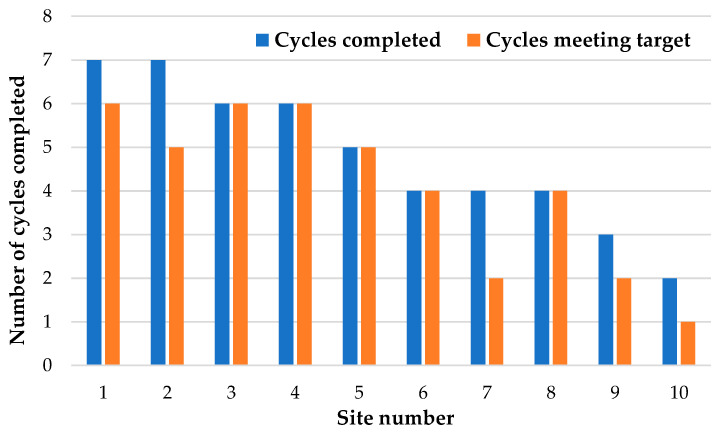
Number of intervention cycles completed by each site.

**Figure 3 children-10-00953-f003:**
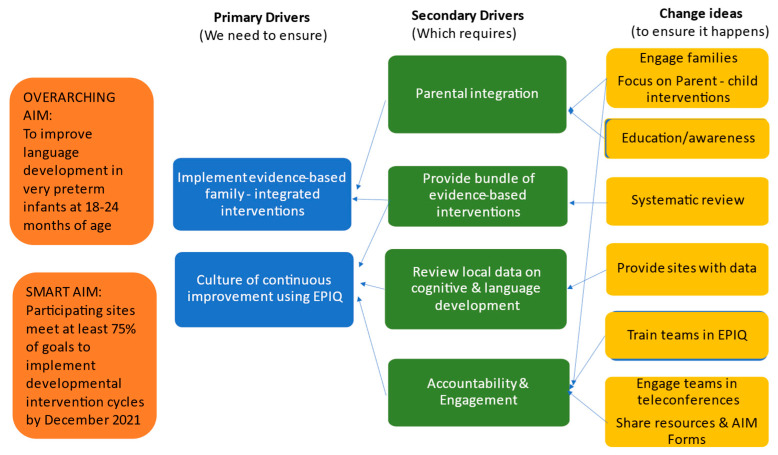
Drivers of success in the Parent-EPIQ model.

**Table 1 children-10-00953-t001:** Intervention Aims and Audit Results by Site.

Site and Size		Aim 1	Aim 2	Aim 3	Aim 4	Aim 5	Aim 6	Aim 7
1 Large	Aim	Improve NFU staff knowledge of importance of parent–infant interaction for language Development with >85% of NFU team attending a workshop and scoring ≥90% on post-test *	Increase knowledge of parental perceptions of early language development for preterm infants and elicit parent input on facilitators/ barriers to participation in language development activities with 10 parent interviews	Create a “Language Passport” brochure for NICU parents with information about how to stimulate early language development in NICU	Validate and improve “Language passport” by receiving feedback from >10 NICU parents	>70% of NICU families report using > 1 language-building activity > three times per week in the NICU	>50% of NICU families report using > 1 language-building activity > three times/week post-NICU discharge	Reduce from 44% to 40% NICU families who report not having been talked to about early language development
	Audit results	15/23 (65%) attended; 15/15 scored >90%	13 interviews; detailed qualitative data obtained	Passport created in two languages AND video created	14 parents provided feedback	86%	24/27 (89%)	Improved greatly but audit not completed
2 Medium	Aim	80% of Parent-EPIQ team to be trained in EPIQ workshop using the planned project	70% of NICU nurses complete education module about reading to babies in NICU	Include nurturing moments (parents reading, talking, singing to babies) to the Voyage to Home parent card as standard of care	100% of NICU single rooms to have a book bags on admission; pamphlet of baby songs created	Increase nurturing moments (reading, talking, singing to babies) to 1 hr/day during 70% of skin-to-skin care	90% of NICU staff trained in SENSE program	Implement family education “Limit Screen time & increase Face time awareness” at 4 mo NFU visit
	Audit results	80%	55%	100%	100%	Increased from 56% to 64%	Achieved and now standard of care.	Became standard of care
3 # Medium	Aim	Increase knowledge of importance of early language stimulation in children born prematurely to 100% of NFU staff by watching an information session	Increase awareness of importance of language stimulation in the NICU by NICU staff from 0% to >70% via written and oral communication	Create a parent-centered language enhancement and reading program in the NICU by giving out a reading bag and verbal coaching; assess % of families receiving a reading bag	Incorporate a NFU formal language enhancement program to >90% of visits for premature children	Develop a lending library in the NICU	Start a NICU volunteer Baby Readers Program in the NICU two times per week to read when parents cannot come	
	Audit results	100% watched; all scored 100% on test	From four surveys, 48/65 (74%) were aware	70% received bag and 88% were talked to	Created; 31/32 (97%) received program	Created; 160 books loaned in 11 months	Readers read for 25–52 hrs/month	
4 # Medium	Aim	Increase knowledge of importance of early language stimulation in children born prematurely to 100% of NFU staff by watching an information session	Improve awareness of the importance of language stimulation in NICU staff and Veteran Parent Program leadership from 0% to 70% via written and oral communication	Create a parent centered language enhancement and reading program in the NICU by giving out a reading bag and verbal coaching to 75%	Incorporate a NFU formal language enhancement program to >90% of visits for premature children	Develop a lending library in the NICU	Start a NICU Baby Readers Program using volunteers	
	Audit results	100% watched; all scored 100% on test	79/90 (88%) aware	83% received book bag/pamphlets and 76% talked to	54/54 (100%) received program	Created with 35 books; mean volunteer hrs/mo = 10.5	Created; mean volunteer hrs/mo = 13	
5 Large	Aim	Improve awareness of language delay among NFU clinical staff by 95%; watching an information session and providing a lesson learned	Provide “The Reading Tree” (provided by the local library) to 50% of families who had not previously received it, at 4 mo to 3-year NFU visits	Increase knowledge and awareness of the importance of communication, language development, and stimulation in premature infants by 85% of NICU nurses attending an education session	Create a list of >15 useful tips to support health care professionals in promoting language development	Publish > 13 language improvement tips in the educational Neonatal Program weekly newsletter		
	Audit results	96% attended or watched video	At 240 visits, 49 (20%) already had the book, 176 (73%) received the book, and 15 (6%) did not	136 (85%) of nurses attended one of six sessions and provided implementation suggestions	31 tips (207%) created	16 tips (123%) published		
6 Large	Aim	Increase % of families watching an approved video on language development at 4-month NFU visit from 0 to 75%	Increase % of families attending music session at 4 or 8-month NFU visit from 0 to 50%	Implement the Read with Me program to 80% of families attending the 4 and 8-month NFU visits	Implement the Read with Me Program to 80% of families during a home visit with a nurse from the Neonatal Transition Team			
	Audit results	First mo: 13/17 (76%) watched; second mo: 9/9 (100%) watched	First mo: 44/49 (90%); second mo: 33/37 (89%); parents rated sessions 4.5/5	28/31 (90%) received book and information	First mo: 18/19 (95%); 80–100%, subsequently			
7 Large	Aim	Improve NFU staff knowledge of importance of parent–infant interaction for language development with >85% of NFU team attending a workshop and scoring ≥90% on post-test *	Increase % of NFU visits post-NICU discharge to 6 months, during which parent–infant interaction is actively discussed to 75%	Increase % of parents aware of mental health issues related to preterm birth from 50 to 90%	Increase % of NFU parents screened for post-partum depression from 0 to 75%			
	Audit results	15/17 (88%); all scored >90%	80%	Did not get buy-in from entire team	Roll-out halted due to COVID-19			
8 Small	Aim	100% of NFU staff attend an interactive workshop on community support and resources for cognitive and language development	Create standardized language development checklists for 4, 8, 12, 18, 24, and 36-month NFU visits	Implement checklists and parent language support information sheets	Book bag (book and library information) gifting at 4 or 8-month visit			
	Audit results	100% attendance	Created	Implemented; estimated 75% uptake by staff	100% of families received a bag			
9 Small	Aim	Improve NFU staff knowledge of importance of parent–infant interaction for language development with >85% of NFU team attending a workshop and scoring ≥90% on post-test *	Increase the number of parental reading behaviors from <1 to ≥3	Increase in % of parents aware of community resources to promote literacy from 0 to 70%				
	Audit results	5/6 (83%); all scored >90%	86%; 100% aware of importance of reading, singing, and talking	80% received pamphlet				
10 Small	Aim	Increase % of parents who reported reading to their babies in NICU from 0 to 80% using a Welcome Baby Bundle	Increase % of NICU staff who engaged parents in discussion about reading, talking, or singing to baby to 80%					
	Audit results	Baseline: 3/11 (27%) After: 7/8 (88%)	Baseline: 4/11 (36%) After: 5/8 (63%)					

* Three sites in the same city collaborated on one cycle. # Two sites in the same city collaborated on all cycles. Hrs—hours; mo—month; NFU—neonatal follow-up program; NICU—neonatal intensive care unit; SENSE—Supporting and Enhancing NICU Sensory Experiences (https://chan.usc.edu/nicu/sense accessed 10 April 2023).

**Table 2 children-10-00953-t002:** Barriers and facilitator questionnaires.

Question	Response
Did you have an adequate knowledge and understanding of the Parent-EPIQ process?	Yes (9/9)
2a. Did someone on your team attend an in-person EPIQ workshop in addition to the Parent-EPIQ training?	Yes (8/9)
2b. If yes, do you think the workshop is essential?	Yes (7/8)
3. Did you have difficulty finding evidence to guide your EPIQ cycles?	No—(5/9) Yes—(3/9) Do not Know—(1/9)
4. Did the COVID-19 pandemic affect Parent-EPIQ?	Yes (9/9)
5. How many parents were on your team at one time?	Mean 1.2
6. How many parents were on your team in total?	Mean 1.8
7. Did you feel supported by your institution in implementing Parent-EPIQ?	Yes (6/9)
8. Did you have a champion?	Yes (7/9)

## Data Availability

Data can be made available by contacting anne.synnes@bcchr.ca.

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
