# Peer review of "Parent-Integrated Interventions to Improve Language Development in Children Born Very Preterm"

_children, 2023, doi:10.3390/children10060953_

Round 1
Reviewer 1 Report
Neurodevelopmental challenges are common in children born very preterm and often persist despite interventions. The study tested the feasibility of using Evidence-based Practice to Improve Quality (EPIQ), a proven quality improvement technique that uses scientific evidence, to improve language abilities in very preterm populations. The study involved 10 neonatal follow-up programs, with the goal of completing 4 intervention cycles and meeting targeted aims in at least 75% of cycles. Systematic reviews were conducted, an online quality improvement educational tool was developed, multidisciplinary teams including parents were created and trained, and sites were virtually supported to implement and audit at least 4 intervention cycles. Eight out of ten sites completed the required cycles, and 85% of the cycles met their aim. COVID-19 was a barrier, but parent involvement, champions, and institutional support facilitated success.
I find the content and results of this study to be interesting. However, I believe that the figures and tables could benefit from being edited and made more professional-looking. This would help to better convey the findings and increase the overall impact of the study.
The variability in the numerosity (i.e., the number of intervention cycles completed) among the 10 neonatal follow-up programs could be considered a potential challenge for the study and should be better addressed. However, the feasibility of the intervention was still achieved, with 8 out of 10 sites completing the required intervention cycles. Despite the variability, the study was able to show that the Evidence-based Practice to Improve Quality (EPIQ) approach was feasible and could improve language abilities in very preterm populations.
The study did not directly compare the asynchronous, video module and teleconference approach used in this study to an in-person approach for teaching the EPIQ process. Therefore, the authors cannot provide a direct comparison between the two methods. However, they did note that the use of teleconference support allowed for frequent communication and problem-solving among the multidisciplinary teams, and that the online educational tool was well-received by the participants. These adaptations were necessary due to the COVID-19 pandemic, which made in-person training difficult or impossible. The authors suggest that future studies could explore the effectiveness of different modes of delivering the EPIQ training, including in-person and virtual methods. However, the fact that this comparison is not presented here makes the study less complete.
Overall, this study presents an interesting approach to improving language abilities in very preterm populations using the Evidence-based Practice to Improve Quality (EPIQ) methodology. While the study showed that the EPIQ approach is feasible and effective, there are some potential areas for improvement. Specifically, the figures and tables could benefit from being edited to improve their professionalism and clarity. Additionally, there was some variability in the numerosity of the intervention among the different neonatal follow-up programs, which could be further discussed. Nonetheless, after minor revisions of the text and major revisions of the figures and tables, this study could be an important contribution to the literature on improving outcomes for preterm infants. Finally, while the asynchronous, video module and teleconference approach for teaching the EPIQ process was well-received, a direct comparison with an in-person approach was not made. Future studies could explore the effectiveness of different modes of delivering the EPIQ training to further refine this methodology.
Author Response
Reviewer 1:
Thank you for your comments which are listed below with our responses in italics.
- I believe that the figures and tables could benefit from being edited and made more professional-looking.
The figures and tables have been reviewed and most have been revised.
- The variability in the numerosity (i.e., the number of intervention cycles completed) among the 10 neonatal follow-up programs could be considered a potential challenge for the study and should be better addressed.
We appreciate your comment. The EPIQ process is designed for different sites to individualize interventions to align with their own strengths and weaknesses and we expect to see variability and numerosity. For readers not familiar with EPIQ, the number of different interventions may appear to be a limitation. The aim of our study was to evaluate the feasibility of using the EPIQ process with it’s inherent variability of intervention cycles rather than identify generalizeable interventions. We have clarified and reinforced this concept in several areas in the text:
In the introduction, on line 81 we have added "site specific": “EPIQ uses local data to direct site specific effective interventions” In section 2.1 Design and setting, in line 117, local has been added “EPIQ uses 3- to 6-month Plan-Do-Study-Act cycles to implement local changes.” The following sentence has been added to line 122 “EPIQ allows each site to individualize the process to their own situation.”
In section 3.4, we have reinforced the concept that interventions are individualized: “Each site used the ten EPIQ steps to understand the improvement opportunities at their own site, how to address them with their own resources and engage their own members to create interventions tailored to their own situation. Sites also had access to scientific evidence from the literature and systematic reviews and the experiences of colleagues at other sites to guide their choices.”
In the discussion we clarified: “As expected, the EPIQ process resulted in different interventions at different sites. The aim was to evaluate the EPIQ process and not to create standardized interventions.”
- Finally, while the asynchronous, video module and teleconference approach for teaching the EPIQ process was well-received, a direct comparison with an in-person approach was not made.
The intent of the study was to describe the Parent-EPIQ process we used in this study. We did not foresee the COVID pandemic and the changes to online teaching that would result. Therefore we did not plan to compare asynchronous video module learning to teleconference learning. We did feel it was necessary to make the reader aware of the changes that have occurred since we carried out our study “Our findings that in person workshops are necessary may not apply to current virtual educational platforms. “
To eliminate the perception that we were comparing asynchronous video modules and teleconference methods of learning we have deleted “virtual/asynchronous learning” on line 365 in the conclusions.
Reviewer 2 Report
Authors aimed to support the benefit of an intervention program on the screening and care of difficulties and delay in development of preterm infants. nevertheless, some limitations should be observed.
in introduction section, authors explained that many interventions programs already exist. Authors should describe the state of art of these interventions and if other project exist aimed to assess the characteristics and benefit of existing interventions. Moreover, the use of parents seemed to be an innovative aspect of the present program. Nevertheless, no mention on the use of parents in other programs is given.
I also suggest to explain that intervention in first months of life were considered. moreover, in data collection and analyses, all aspects described in results should be explained.
in method section, authors should explain why only 10 of 26 program are considered. did they no respect some criteria? or are they somehow different? is this study preliminary?
Author Response
Thank you for your comments which are listed below with our responses in italics.
- In introduction section, authors explained that many interventions programs already exist. Authors should describe the state of art of these interventions and if other project exist aimed to assess the characteristics and benefit of existing interventions.
In the introduction we stated that randomized control trials have been published “ Systematic reviews and meta-reviews [ 9-12] have identified interventions that improve infant developmental outcomes in children born preterm” but we didn’t say they have been implemented. In Canada we know from informal discussions with colleagues that these interventions have not been implemented. We have therefore added the following in line 89 of the introduction: “To the best of our knowledge the interventions described in the systematic reviews to improve neurodevelopmental outcomes have not been implemented in Canadian neonatal follow up programs.” We are not aware of other studies that have assessed whether these interventions have been implemented and if so the characteristics and benefits. To describe the state of the art of all these interventions would probably require 1,000 words and distract the reader from the aim or our study so we felt it was more appropriate to provide the appropriate references.
- Moreover, the use of parents seemed to be an innovative aspect of the present program. Nevertheless, no mention on the use of parents in other programs is given.
Thank you for your comment. I was not aware of any similar programs. I have searched PubMed today and no results were found. We have therefore acknowledged the innovative aspect and lack of previously report studies: “Involving parents in new health care interventions is innovative, has not been previously reported and our study has shown this to be both possible and beneficial.”
- I also suggest to explain that intervention in first months of life were considered.
Thank you for this suggestion which was not previously specified. We have added” Any evidence based change to improve language and / or cognitive outcomes prior to 18 months of age in the preterm population assessed by the site and within the sphere of influence of the Parent-EPIQ team were considered to be within the scope of this study.” under Methods on line 146.
- moreover, in data collection and analyses, all aspects described in results should be explained.
Thank you for prompting us to review that the descriptions of the data collection and analyses are complete and that all aspects described in results are included under data collection and analyses.
Description of intervention cycles and Table 1: The data collection which was described under Design and setting is now explicitly mentioned under Data collection and analyses: “The descriptive variables were obtained from the electronically submitted EPIQ aim forms, process indicators and audit results and confirmed with site investigators.
The calculation of completed cycles and successful completion of cycles was previously described: “The two feasibility criteria were defined as 1) 70% of participating sites complete at least 4 interventions cycles during the study and 2) 75% of all intervention cycles meet or surpass their target goals. At each site, for each EPIQ cycle an aim form with specific measurable goals was completed and an audit performed to address whether targets were met. The number of cycles completed for each site was recorded and the percentage of sites completing at least 4 intervention cycles calculated. The percent of audit results that met or surpassed target goals was calculated for all cycles at all sites.”
The data collection used to assess the reasons for not meeting the targeted number of cycles has been added: “Site investigators who were unable to meet the target were asked to identify the factor(s) which hindered the completion of at least 4 cycles.”
The data collection for the barriers and facilitators questionnaire was moved from design and setting to data collection and analyses: “Using the conceptual framework of Cane et al [32], a survey was created to assess the barriers and facilitators to implementation of Parent-EPIQ and circulated to participating site investigators by email after completion of all the intervention cycles in June 2022.”
- in method section, authors should explain why only 10 of 26 program are considered. did they no respect some criteria? or are they somehow different? is this study preliminary?
As a research study, program participation was voluntary. Site investigators had to commit to obtaining research ethics board approval and spend time to follow the research protocol. All 26 programs were notified about the research study when the funding grant was being prepared and asked whether they were interested in participating. Potential site investigators provide letters of intent. After funding was received, site investigators needed to obtain research ethics board approval.
Canadian neonatal follow up programs vary in size, whether they belong to an academic hospital or not, whether medical directors are expected to participate in research or not and their ability to participate in research studies. This, the first study of its kind, evaluated feasibility at sites willing and able to participate and can therefore be considered a large pilot study.
The follow has been added under discussion to further explain why some sites may not have participated: “Though all 26 Canadian neonatal follow up programs were invited to join the study, participation was voluntary and 16 sites, mostly smaller ones with less resources, elected not to. We had previously identified the considerable variability in the size and available resources of Canadian neonatal follow up programs [42] and their ability and academic expectations to participate in research. The experiences learned in this study will help sites evaluate whether, and what would be needed to participate in future implementation projects.” We also added reference 42 which describes the variability in Canadian neonatal follow up programs.